# Novel Cecropin-4 Derived Peptides against Methicillin-Resistant *Staphylococcus aureus*

**DOI:** 10.3390/antibiotics10010036

**Published:** 2021-01-01

**Authors:** Jian Peng, Biswajit Mishra, Rajamohammed Khader, LewisOscar Felix, Eleftherios Mylonakis

**Affiliations:** 1Infectious Diseases Division, Rhode Island Hospital, Warren Alpert Medical School of Brown University, Providence, RI 02903, USA; jian_peng@brown.edu (J.P.); biswajit_mishra@brown.edu (B.M.); rajamohammed_khader@brown.edu (R.K.); lewis_oscar_felix_raj_lucas@brown.edu (L.F.); 2Immune Cells and Antibody Engineering Research Center of Guizhou Province, Key Laboratory of Biology and Medical Engineering, School of Biology and Engineering/School of Basic Medical Sciences, Guizhou Medical University, Guiyang 550025, China

**Keywords:** cecropin, antimicrobial peptides, biofilm, persisters, *Staphylococcus aureus*

## Abstract

Increasing microbial resistance, coupled with a lack of new antimicrobial discovery, has led researchers to refocus on antimicrobial peptides (AMPs) as novel therapeutic candidates. Significantly, the less toxic cecropins have gained widespread attention for potential antibacterial agent development. However, the narrow activity spectrum and long sequence remain the primary limitations of this approach. In this study, we truncated and modified cecropin 4 (41 amino acids) by varying the charge and hydrophobicity balance to obtain smaller AMPs. The derivative peptide C18 (16 amino acids) demonstrated high antibacterial activity against Gram-negative and Gram-positive bacteria, as well as yeasts. Moreover, C18 demonstrated a minimal inhibitory concentration (MIC) of 4 µg/mL against the methicillin-resistant *Staphylococcus aureus* (MRSA) and showed synergy with daptomycin with a fractional inhibition concentration index (FICI) value of 0.313. Similar to traditional cecropins, C18 altered the membrane potential, increased fluidity, and caused membrane breakage at 32 µg/mL. Importantly, C18 eliminated 99% persisters at 10 × MIC within 20 min and reduced the biofilm adherence by ~40% and 35% at 32 and 16 µg/mL. Besides, C18 possessed a strong binding ability with DNA at 7.8 μM and down-regulated the expression of virulence factor genes like agrA, fnb-A, and clf-1 by more than 5-fold (*p* < 0.05). Interestingly, in the *Galleria mellonella* model, C18 rescued more than 80% of larva infected with the MRSA throughout 120-h post-infection at a single dose of 8 mg/kg (*p* < 0.05). In conclusion, this study provides a reference for the transformation of cecropin to derive small peptides and presents C18 as an attractive therapeutic candidate to be developed to treat severe MRSA infections.

## 1. Introduction

Antimicrobial resistance (AMR) has become a global crisis that presents a substantial public health threat [1]. Antibiotic-resistant infections are responsible for 700,000 deaths annually worldwide, and the number is projected to rise to 10 million every year by 2050 [1]. More recently, secondary infections after COVID-19, such as ventilator-associated pneumonia and bloodstream and urinary tract infections, are posing a burden on antibiotic usage and are expected to increase AMR [2,3]. According to the antibiotic resistance threats report of Centers for Disease Control and Prevention (CDC), in 2017, there were nearly 120,000 *Staphylococcus aureus* bloodstream infections and 20,000 deaths in the United States [4]. At present, vancomycin is the primary drug of choice for methicillin-resistant *S. aureus* (MRSA) infections [5]. However, its toxicity, low tissue penetration, and the emergence of drug-resistance limit its clinical application [6,7]. Thus, the situation necessitates developing a new class of antibacterial agents that would outsmart the AMR infection.

Antimicrobial peptides (AMPs) have received particular attention for the development of novel antibacterial agents. Significant families of AMPs, including cathelicidin, defensins, dermaceptins, temporins, cecropins, magainin, protegrin, etc., have been discovered and characterized extensively from the major phylum from animals, plants, fungi, and bacterial for their antimicrobial properties [8,9]. Most of the AMPs adopt an amphipathic α-helical structure and have a membrane mimetic mechanism of action. AMPs possess a short killing time, and hence it is challenging for the pathogen to develop resistance. Their favorable toxicity profile, combined with activity against multidrug-resistant strains, biofilms, and bacterial tolerance, makes AMPs excellent candidates for drug development [10]. For example, short α-helical AMPs have been characterized with excellent in vivo potencies. Ω76 was highly effective in a mouse model of carbapenem and tigecycline-resistant *Acinetobacter baumannii* infection [11]. Magainin II-based stapled AMPs were excellent in reducing colistin-resistant *A. baumannii* infections in a peritonitis/sepsis mouse model with no renal toxicity [12]. Another class of arginine and tryptophan-rich peptides was effective in reducing systemic *S. aureus* burdens in most of the vital organs in intraperitoneal and intravenous injection modes [13].

Previously, our group reported a new member of the cecropin family named cecropin4 (Cec4) with significant potential to be developed into peptide antibacterial agents [14,15]. However, with a long 41 amino acid chain and a narrow antibacterial spectrum, the commercial development of Cec4 remains challenging. Thus, in the present study, we attempt to design short and potent Cec4-derived AMPs. We obtained shorter peptides by intercepting the parent peptide’s fragments and by the selective substitution of hydrophobic amino acids. Interestingly, a derivative peptide, C18, composed of only 16 amino acids, was found to have broader antibacterial effects, including against drug-resistant *S. aureus* strains. We performed experiments focusing on the development of C18 as a potential antibacterial candidate against MRSA infection and explored the mechanism of action, synergistic activities, anti-biofilm potential, and the toxicity profile of the peptide. Besides this, we also conducted pilot in vivo efficacy studies evaluating the use of C18 peptide against *S. aureus* in a *Galleria mellonella* (wax moth) model.

## 2. Results

### 2.1. Designing of Derived Peptides and Evaluation for Antimicrobial Activity

The sequence of Cec4 is 41 amino acids long, and our first priority was to design peptides of shorter length for further antibacterial agent development studies. In total, 24 peptides were designed by modifying the length and amino acids of the parent peptide Cec4 (Figure 1A, Appendix A). The figures reveal that most of hydrophobic amino acids of C9 and C18 are concentrated on one side of the helix, with polar or hydrophilic amino acids on the other, while the trend of Cec4 was not significant (Figure 1B–D). Besides, the helical wheels of other synthetic peptides were shown in Appendix A. Since our initial assumptions presumed the presence of both charged and hydrophobic amino acids on the primary sequence of Cec4, we selected the N-terminal helix to design the short peptides. Typically, the N-terminal helix of most cecropins starts from the Lys3 residue [16]. We started the design from Leu3 and continued to Gln23 (the numbering of amino acids is based on the parent Cec4 sequence), called peptide C1, which can form a complete helix (Figure 1A). Peptide C1 demonstrated no activity against any of the *Enterococcus faecium*, *Staphylococcus aureus*, *Klebsiella* spp., *Acinetobacter baumannii*, *Pseudomonas aeruginosa*, and *Enterobacter* spp. (ESKAPE) pathogens or *C. albicans* (Table 1). However, deleting the last 5 C-terminal residues, which were neither charged nor hydrophobic, did not make a peptide with better antimicrobial activities with MIC value ≥128 µg/mL against the ESKAPE pathogens and *C. albicans* (peptide C2) (Table 1).

The primary sequence of Cec4 has sufficient quantities of charged amino acids (2 Arg and 4 Lys) needed to bind to the bacterial membrane but does not possess antimicrobial activities against most of the ESKAPE pathogens; hence, we directed our attention towards increasing the hydrophobic contents. For this, we sequentially replaced the neutral, less hydrophobic amino acids Thr and Lys with higher hydrophobic amino acids like Trp and Leu. Indeed, the replacement of neutral Thr17 with Trp increased the potency of the peptide C3. It was found to have an increased minimal inhibitory concentration (MIC) value of 32 µg/mL against *A. baumannii* and was also broad spectrum with the MIC value of 128 µg/mL against *S. aureus* strain MW2 and *C. albicans*. Interestingly, deleting a further 4 amino acids (approximately 1 α-helical turn) made the peptide (peptide C4) inactive.

We focused on peptide C3 and increased the hydrophobic contents even more by replacing the charged amino acids. Two charged clusters of peptide C3 consisting of two Lys units each (Lys4/5 and Lys8/9) were selected for hydrophobic replacement. The substitution in Lys5 (peptide C5) made the broad peptide spectrum with MIC ranges from 32–128 µg/mL against all ESKAPE pathogens, while a double replacement involving the Lys9 made the peptide ineffective (peptide C6). The result indicated that the peptide C7 became inactive against all tested strains, and C8 became inactive against the ESKAPE pathogens and *C. albicans*, except for *S. aureus* MW2 (Table 1). Therefore, when giving the peptides C7 and C8 higher hydrophobicity, the deletion of a further 1 turn helix (4 amino acid residues) was detrimental to their antimicrobial activity against most of the ESKAPE pathogens and *C. albicans*.

Interestingly, the replacement of the set of Lys at residue 4 (peptide C9) made the peptide very effective. The MIC of peptide C9 was 16 µg/mL against *A. baumannii* and 64 µg/mL towards *S. aureus* MW2 and *P. aeruginosa*, suggesting Lys4 might be less involved in the membrane binding. A double displacement variant (peptide C10) was twice as active as peptide C9 MIC against *A. baumannii*. Further deletion of one turn (4 amino acids) in peptides C11 and C12 made the peptide ineffective when faced with MIC ranges from 64 to >128 µg/mL (Table 1). Hence, we considered peptide C9 the shortest (16 amino acid residues) peptide suitable to be further optimized.

A series of 16 amino acid long peptides was designed primarily based on increased charge/hydrophobic contents based on the peptide C9 (Table 1). Hydrophobic amino acids facilitate the formation of pores in the cell membrane. To improve the overall hydrophobicity of the peptide, two Gly at positions 7 and 14 were the most replaceable amino acids because of their lower hydrophobicity. Simultaneously replacing Gly7 and Gly14 with Trp made the peptide very effective. The peptides C13 and C14 had similar MIC values against ESKAPE pathogens and *C. albicans*, while their double substitution was not as effective (peptide C15) (Table 1).

Reverse substitution of Lys15 to Trp (peptide C16) was found to be more active against *S. aureus*, although the MIC was higher by twofold against Gram-negative *K. pneumoniae* and *A. baumannii*. Peptide C17 obtained by reverse substitution of Gly7 with Leu in peptide C9 made the peptide less active than C16 (Table 1). However, the substitution of Gly14 in C9 with Leu (Peptide C18) increased the MIC against *S. aureus* and other ESKAPE pathogens. Further substitution of Gly7 to Leu (peptide C19) or hydrophobic Trp with changed (peptide C20), even double substitution of neutral and charged amino acids (peptide C21) and their reverse replacement (peptide C22) made the peptide less effective against *A. baumannii* with MIC values of > 128 µg/mL (Table 1).

It is worth noting that replacing Trp at positions 7 and 14 made the peptide inactive with relation to MICs > 128 µg/mL (peptide C23), even if a charge reversal at the 15^th^ position did not add any potency (peptide C24), and its MIC values against all ESKAPE pathogens and *C. albicans* were > 64 µg/mL. Overall, the MIC of C18 was found to be best against *S. aureus* at 4 µg/mL, 16 µg/mL towards *K. pneumoniae*, *A. baumannii*, *E. aerogenes*, and *C. albicans* (Table 1).

### 2.2. Salts and Serum Effect and Synergy with Clinical Antibacterial Agents

The MIC tests showed that the antimicrobial activities of C13, C18, C22, and vancomycin were not impeded in the presence of salts (2 mM Ca^2+^ or 150 mM NaCl). However, the MIC values of C13, C18, and C22 increased to 128 µg/mL in a medium containing a physiological amount (5%) of human serum, while the MIC value of vancomycin was not changed (Table 2). We also found that when peptide C18 was combined with daptomycin against *S. aureus* strain MW2, the FICI value was 0.313 (FIC < 0.5), which indicated a synergistic effect. When C18 was combined with vancomycin, gentamicin, oxacillin, or ciprofloxacin against *S. aureus* strain MW2, the FICI values were, respectively, 0.625, 1.25, 0.75, and 0.75, and they showed an additive effect (Table 3).

### 2.3. Mechanism of Action

To elucidate the mechanism of action of the designed AMPs, we studied the interaction of the peptide with the bacterial and model membranes (Figure 2). The initial interaction of the peptides in contact with the *S. aureus* membrane was performed with a fluorescent-based DISC3 (5) dye. The lipophilic potentiometric dye changes its fluorescence upon the change in transmembrane potential [17]. Interestingly, all the peptides induced a change in the bacterial transmembrane potential. At 32 µg/mL, peptide C18 demonstrated increased fluorescence change owing to its low MIC values and high net charge (Figure 2A). The Triton X-100 was used as a positive control, and bacteria treated with water acted as a negative control.

We also evaluated the propidium iodide fluorescence of *S. aureus* MW2 cells after treating them with C18 at 32 µg/mL. Live bacterial cells have intact membranes and are *impermeable* by dyes such as *PI* (*Propidium iodide*) [18,19]. However, the interaction of the AMPs with the bacterial membrane results in membrane damage, and allows the dye to enter cells, integrate with the DNA and increase the fluorescence. Peptides C13, C16, C18, and C22 all demonstrated increased fluorescence intensities compared to control after 1 h of incubation with the bacterial membrane and dye (Figure 2B), indicating dye entry into the bacterial cells. The ability to induce PI fluorescence was in the order of C22, C18, C16, and C13, respectively. As anticipated, the bacterial control and the cell wall binding antibacterial agent, vancomycin, were unable to induce any fluorescence.

We also performed dye leakage experiments to further understand the membrane interactive nature of the peptides. A dye, rhodamine-BF, entrapped in liposomes (DOPC: Cholesterol, 55:40) was subjected to the action of the peptides. Interestingly, at the end of 1 h of incubation, the fluorescence of the peptides was recorded in the order of C18, C13, C16, and C22 (Figure 2C). The bacterial control and the vancomycin demonstrated the lowest, and the Triton X-100 control the highest fluorescence. Figure 2D shows the killing rate of C18 against MRSA MW2 persisters after being treated with peptide C18 at 1–10 times MIC for 4 h at 37 °C. At 1 × MIC, peptide C18 eliminated most of the inoculum within 240 min, and it eliminated 99% of bacteria within 20 min at 10 × MIC.

In the next series of experiments, we used Laurdan dye to measure the fluidity change of bacterial membranes. The fluorescence emission pattern of dyes depends on the number of water molecules between lipid head groups, so changes in lipid head group density and fatty acid chain flexibility can be estimated [20]. A significant decrease in membrane fluidity was obtained by using benzyl alcohol (50 mM) as a membrane fluidizer. Generalized polarization (GP) values at 32 µg/mL were significant for C13, C16, and C18, indicating the change in a microfluidic environment (Figure 2E).

Polymyxin B has been shown to bind to DNA [21]. Due to the amphiphilic structure of polymyxin B, it binds to DNA non-covalently through electrostatic attraction, groove binding, and insertion. In our assays, polymyxin B was able to bind DNA significantly at 31.2 μM (37.1 µg/mL) and has a concentration-dependent effect. Similarly, C18 exhibited significant binding at 7.8 μM (21.6 µg/mL), leaving almost all the DNA in the well. Higher concentrations (31.2, 125, and 500 μM) also indicated that DNA was still present in the respective wells (Figure 2F). The result showed that C18 has a stronger ability to bind to DNA compared with polymixin B.

### 2.4. Anti-Biofilm Effects

The antibiofilm potential of the peptides was tested at different stages of biofilm formation. In an initial series of experiments, the biomass attachment on the polypropylene surface was measured using high-density *S. aureus* cultures. Interestingly, peptide C13 inhibited the adherence of MRSA cells. At the highest concentration of 32 µg/mL, it reduced cell adherence by ~30%. Peptide C16 was unable to reduce any adherence even at its highest concentration. Excitingly, the peptides C18 and C22 were effective even at 16 µg/mL with similar efficacies and reduced the adherence by ~40% and 35% at 32 and 16 µg/mL, respectively (Figure 3A–D).

Peptides were also tested against MRSA biofilm. Peptide C13 was unable to exert any favorable effect at any of the concentrations tested. However, the peptides C16, C18, and C22 reduced the number of live cells during incubation. Peptide C16 demonstrated efficacy at 32 µg/mL and resulted in a ~90% reduction of biofilm. However, C18 completely eradicated any live cells at 32 µg/mL, and peptide C22 was active at 16 µg/mL. Moreover, C18 reduced the biofilms by ~60% at 32 µg/mL and ~30% at 16 µg/mL, respectively (Figure 3E–H).

### 2.5. The Effect of C18 on the Regulation of the Virulence Genes in S. aureus

The virulence genes are known to be responsible for the emergence of antibacterial agent-resistant strains of *S. aureus*. Targeting these virulence genes will be a potential strategy for combating drug-resistant *S. aureus* strains [22]. The effect of the C18 on the regulation of the *S. aureus* virulence genes agrA, Spa, fnb-A, fnb-B, clf-1, and srrA was also tested. Interestingly, in the presence of sub-MIC levels of C18 peptide, key virulence genes were down-regulated by more than fivefold. These include the accessory gene regulator A (agrA), fibronectin-binding protein A (fnb-A), and cytokine-like factor 1 (clf-1) (*p* < 0.05) (Figure 3I).

### 2.6. Hemolytic Activity and Cytotoxicity

To evaluate for mammalian toxicity, we studied the hemolytic potential of peptides C13, C16, C18, and C22. This series of hemolysis experiments was performed using 2% human red blood cells (Figure 4A). Along with peptide C18, we included peptides C13, C16, and C22 as controls. The HL_50_ (50% of human red blood cells were lysed) of the C18 was found to be 16 µg/mL, which is 4 times its MIC concentration against MRSA MW2. The least hydrophobic peptide C13 demonstrated the lowest HL_50_ of 48 µg/mL. In comparison, the highest hydrophobic peptide, C22, demonstrated an HL_50_ of 4 µg/mL. C16 and C18 were found to cause similar hemoglobin release owing to similar hydrophobicity. Interestingly, the LD_50_ (lethal dose of a drug for 50% of the population) of all the tested peptides towards the liver-derived HepG2 cell lines were found to be > 64 µg/mL. More than 90% of the cells were found to be intact. As anticipated, peptide C22 was the most toxic, reducing the live-cell contents by 20% at 64 µg/mL (Figure 4B).

### 2.7. Galleria Mellonella Assays

C18 peptide was used to investigate the toxicity and utilized to access the virulence of bacterial pathogens using *G. mellonella*. Toxicity was tested using a series of increasing doses (4, 8, 16, 32 mg/kg). As shown in Figure 5A, more than 70% of larvae survived the toxicity until 120 h. Due to the fact that C18 has a degree of hemolysis, at high concentrations (32 mg/kg), it also shows a certain influence on the survival of *G. mellonella*. In addition, after *G. mellonella* larvae were infected with MRSA, we treated the larvae with 4 different concentrations and all concentrations resulted in improved survival (*p* = 0.025) (Figure 5B). Interestingly, the group treated with 4 mg/kg had 80% survival at 120 h post-infection, while in the 32 mg/kg group the survival at rate 120 h was 70%. Even though this difference is not statistically significant (*p* = 0.09), it is probably due to toxicity and provides us with some first insights on the window between efficacy and toxicity. Importantly, these results indicate that peptide C18 is significantly efficacious in the *G. mellonella* model.

## 3. Discussion

Infections caused by MRSA impose a significant burden on healthcare systems [1]. Most cecropins are 31–40 amino acids and have broad activity against a wide range of pathogens [23]. It is suggested that these peptides are promising antibacterial therapeutic candidates because of their low toxicity against mammalian cells and anti-inflammatory activity [23]. In this study, the smaller peptide C18 (16 amino acids) derived from cecropins demonstrated a MIC of 4 µg/mL against the MRSA even in the presence of high salt concentrations and showed synergism with daptomycin. Similar to traditional cecropins, C18 altered the membrane potential, increased fluidity and caused membrane breakage. Importantly, peptide C18 eliminated persisters and MRSA biofilms by acting at the initial adhesion and biofilm establishment stages by downregulating virulence factor genes like agrA, fnb-A, and clf-1. In addition, peptide C18 rescued more than 80% of *G. mellonella* larva infected with the *S. aureus*.

Natural cecropin and cecropin-like peptides generally exhibit antibacterial activity against different bacteria; however, the vast majority of these peptides demonstrate no activity against *S. aureus* [23]. Members of our group have previously shown that some of the 11 cecropin molecules of the housefly demonstrate antibacterial ability against Gram-negative bacteria, including Cec4, with excellent effects against *A. baumanni* [14,15]. In general, the N-terminus of cecropin forms a helix structure, which plays an essential role in its antibacterial function [24]. Therefore, we chose the N-terminal helical region to design the short peptide and further change the composition of different charge and hydrophobicity of the amino acids.

Through sequence truncation, we found that the number of amino acids should not be less than 16; otherwise, the antibacterial activity was lost (C1, C2, and C4). Then, we used peptide C3 as a template to increase the hydrophobicity of the peptide chain through amino acid substitution by replacing the charged amino acids with a hydrophobic amino acid. The cluster composed of two Lys units in the sequence may be related to the ability of bacterial membrane binding. Interestingly, peptide C9 (the Lys replacement at residue 2) showed the MIC against *A. baumannii* was 16 µg/mL, and the double displacement mutation (peptide C10) showed stronger antibacterial ability against *S. aureus* than peptide C9. Therefore, considering its antibacterial activity against *A. baumannii* at 16 µg/mL in our studies, we consider C9 to be the shortest (16 amino acid residues) peptide among the peptides we made.

A combination of high hydrophobicity and low cationicity is crucial for maintaining the antimicrobial ability of the peptide [25]. Since we have already substituted the charged amino acids with hydrophobic amino acids in the first round, we believe that proceeding in a similar fashion will lead to less charge and break the balance between the charge/hydrophobic peptides. Hence, to increase or decrease the hydrophobicity of the peptide sequence, we altered residues Trp, Gly and Leu. We found that Lys15 reversal replacement of Trp (C16 peptide) was more active against *S. aureus*, while Leu (peptide C18) replacement increased MIC against *S. aureus* and other ESKAPE pathogens. Although some synthetic cecropin-analogs against *S. aureus* have been reported, most of them were obtained by stitching together sequences of different antimicrobial peptides, such as CA-MA or CA-LL37 [26,27]. Therefore, the new antimicrobial peptide C18 is smaller in molecular weight and extends its antimicrobial spectrum to Gram-positive bacteria and *C. albicans*.

Most cationic AMPs are sensitive to salt. For example, with the increase of salt concentration, defensin’s antibacterial activity decreases [28], while the human antibacterial peptide LL-37 also shows decreased activity after binding with human serum [29]. However, our results showed that the presence of different salts did not affect the antibacterial activity of C18 (Table 2). Due to the increase in hydrophobicity, the small peptide C18 was more likely to bind to macromolecules in serum, and MIC was increased. Interestingly, C18 was less toxic to liver-derived HepG2 cells, and LD_50_ was found to be >64 µg/mL. In this condition, surface administration in the future may reduce the effect of serum on C18. Combinations of antibacterial agents are widely used to treat infections caused by *S. aureus* and *P. aeruginosa*, as well as tuberculosis and malaria [30]. Interestingly, C18 showed a synergistic effect with daptomycin in vitro, so C18 combined with daptomycin.

The cecropin peptides bind to the lipid bilayer on the membrane surface of the bacteria [31]. Then, the polar residue of the AMPs interacts with the lipid phosphoric acid, and the non-polar side chain drills holes in the hydrophobic core of the membrane, ultimately leading to bacterial lysis [32]. We found that C18, like traditional cecropin peptides, can change the membrane potential, disturb the fluidity, and destroy the bacteria membrane structure, leading to bacterial death (Figure 2). The presence of a large number of tolerant cells and persisters in *S. aureus* stationary suspension cultures and biofilms are the cause of chronic and relapsing infections, such as pneumonia, heart valve infections, osteomyelitis, and prosthetic implant infection [33]. Our results also indicate that C18 could eliminate MRSA persisters. These results reflect the fact that C18 could eliminate *S. aureus* in different growth states. Virulence factors produced by bacteria are a significant cause of severe infection and many natural products inhibit enterotoxins, type III secretion systems (T3SS), and biofilm. The Quorum sensing (QS) and pathogenicity of *S. aureus* is regulated by a number of virulence genes like agrA, Spa, fnb-A, fnb-B, clf-1 and srrA. In *S. aureus*, the QS system regulates the major virulence factors with autoinducer peptides as signals and activates the expression of exoenzymes and exotoxins [22,34]. The upregulation of agrA is responsible for soft tissue infection in 90% of *S. aureus* infection cases. In addition, fnb-A is responsible for the adhesion to, colonization of, and invasion of the host cell [35,36], while clf-1 plays an essential role in the adherence of *S. aureus* to fibrinogen and fibrin. Targeting these virulence genes could provide an interesting strategy for combating drug-resistant *S. aureus* strains [37]. C18 down-regulate the expression of agrA, fnb-A, and clf-1, thus interfering with the attachment or adherence factors of *S. aureus*.

In summary, we developed an antimicrobial peptide with a broader antibacterial spectrum and a smaller molecular weight through a series of modifications of the 41-amino acid cecropin 4. The enhanced efficacy of peptide C18 helps us to understand the relationship between functional structure and activity of α-helix peptides and provide references for future peptide synthesis. The N-terminal helix of most of the cecropins is their main functional domain and keeping 4 α-helical turns (16AA) is necessary to maintain its antimicrobial properties. Reducing the positive charge of peptides and increasing their hydrophobicity contributes to broadening their antibacterial spectrum, especially against MRSA. Peptide C18 affected the cell membrane structure of *S. aureus* under different growth conditions, thus leading to bacterial lysis. Besides, the inclusion of nucleic acid-based RNA sequencing studies and the study of C18 in higher animal models will shed more light on its in vivo potential, toxicity and mechanism of action. Our study provides a reference for the transformation of cecropin and provides C18 as an attractive therapeutic candidate for treating MRSA infections.

## 4. Materials and Methods

### 4.1. Bacterial Strains and Growth Conditions

The bacterial strains used in this study are commonly known as the *Enterococcus faecium*, *Staphylococcus aureus*, *Klebsiella* spp., *Acinetobacter baumannii*, *Pseudomonas aeruginosa*, and *Enterobacter* spp. (ESKAPE). The members are known to exhibit AMR, virulence, being involved in human-centric infections [38]. *E. faecium* E007, *S. aureus* ATCC29231, *S. aureus* MW2, *K. pneumonia* WGLW2, *A. baumannii* ATCC17978, *P. aeruginosa* PA14, *E. aerogenes* ATCC13048, and *Candida albicans* ATCC10231. The *E. faecium* was grown in brain-heart infusion (BHI) broth (BD, Franklin Lakes, NJ, USA). *S. aureus* and the 4 Gram-negative species, *K. pneumoniae*, *A. baumannii*, *P. aeruginosa*, and *E. aerogenes* were grown in tryptic soy broth (TSB) (BD, Franklin Lakes, NJ, USA), and *C. albicans* was grown in Sabouraud dextrose broth (SDB) medium (BD, Franklin Lakes, NJ, USA). All cultures were performed using an incubator shaker at 37 °C, 200 rpm. Multi-sequence alignment of Cec4 and derived peptides were performed using ClustalW (http://embnet.vital-it.ch/software/ClustalW.html), and the helical wheels were constructed with NetWheels (http://lbqp.unb.br/NetWheels/). Cec4 derived peptides with a purity of >95% (evaluated using High-Performance Liquid Chromatography, HPLC) were synthesized by solid-phase chemistry by Shanghai Gill Biochemical Co., LTD (Appendix A). The peptides were dissolved in double-distilled water (ddH_2_O) to 10 mg/mL stocks and stored at −80 ℃ for further analysis.

### 4.2. Minimal Inhibitory Concentration (MIC) Assay

Broth microdilution assays determined the MICs of peptides following the method described by the Clinical and Laboratory Standards Institute [39]. In short, for each peptide, serial dilutions (50 µL) in duplicates were made in Mueller-Hinton broth (BD, Franklin Lakes, NJ, USA) in a 96-well plate (Cat No. 3595, Corning, NY, USA). Then, each well of the plates containing peptides was laid by 50 µL of logarithmic-phase bacteria at 5 × 10^5^ CFU/mL, and the final concentration ranged from 1–128 μg/mL for each peptide. After bacterial addition, the plates were incubated at 37°C for 18 h. MIC determination was performed by reading the OD_600_ using a SpectraMax M2 spectrophotometer (Molecular Devices, CA, USA) and was referred to as the lowest peptide concentration suppressed bacterial growth.

### 4.3. Antimicrobial Activity in the Presence of Salts and Serum

To investigate each peptide’s activity in high salt concentrations or human serum, the MIC was again tested as described in the previous paragraph of the experimental method. Logarithmic-phase MRSA MW2 cells were incubated with different concentrations of C18 and vancomycin in MHB with NaCl (150 mM), CaCl_2_ (2 mM), or human serum (5%). After culturing at 37 °C for 18 h, the minimum concentration of peptides or antibacterial agents that can completely inhibit the visible growth of bacteria was determined as MIC. All MIC assays were conducted in duplicate, with each experiment replicated twice.

### 4.4. Synergy with Clinical Antibacterial Agents

The checkerboard assay was used to evaluate the synergy of C18 with conventional antibacterial agents against *S. aureus* MW2 [40]. Briefly, an 8 × 8 matrix was formed in a 96-well microtiter petri dish with two-fold serial dilutions of C18 combined with each conventional antibacterial agent. The fractional inhibition concentration index (FICI) was calculated: FICI = MIC of compound A in combination / MIC of compound A alone +MIC of compound B in combination / MIC of compound B alone. Synergy, FICI ≤ 0.5; additive effect, 0.5 < FICI ≤ 4; antagonism, FICI > 4.

### 4.5. Membrane Depolarization

Bacterial membrane potential measurements were done following an established protocol with minor modifications [19]. Exponential phase *S. aureus* MW2 bacteria culture was prepared in fresh MHB media, washed 3 times in PBS (Phosphate Buffer Saline), and resuspended in a double PBS volume. The bacterial cells were energized by adding 25 mM of glucose for 15 min at 37 °C, and the transport rates of the cells were increased. After that, 5.0 μM of DISC_3_ (3, 3’-Dipropylthiadicarbocyanine iodide) (5) (Molecular Probes/Thermo Fisher Scientific, ON, Canada) was added to the bacterial culture, and 50 µl was distributed in a 96-well black, clear-bottom plate (Cat no. 3904, Corning, NY, USA). The plate’s fluorescence was recorded for 20 min at excitation and emission wavelengths of 610 nm and 660 nm, respectively, until the baseline got stabilized. Then, 50 µl of serially diluted peptide solutions were added, and the fluorescence was recorded for another 40 min. Triton X-100 (1%) was used as a positive control.

### 4.6. Propidium Iodide-Based Membrane Permeability

Fluorescence-based bacterial permeation assay was used to follow the membrane-oriented interactions of the designed peptides. The propidium iodide probe is generally used to establish a pore formation mechanism of AMPs. In short, *S. aureus* MW2 was inoculated in MHB and grown overnight at 37 °C overnight with shaking at 220 rpm. A fresh inoculation was done the next day in the same media to achieve exponential growth. The culture was then washed with PBS 3 times and adjusted to OD_600_ ~0.3 in PBS. We added 48 μL of this culture to a 96-well black clear-bottom plate (Cat no. 3904, Corning, NY, USA) containing 50 μL of serially diluted 2 × peptide concentrations, and 2 μL of propidium iodide (20 μM) was added per well. Fluorescence was measured by SpectraMax M2 (Molecular Devices, CA, USA) with an excitation wavelength of 584 nm and an emission wavelength of 620 nm after 1 h at room temperature.

### 4.7. Laurdan Based Membrane Fluidity Assay

The peptide-induced bacterial membrane fluidity was measured by following an established protocol [41] with minor modifications. In short, the exponential phase of *S. aureus* MW2 bacteria was regrown from an overnight culture in fresh MHB media. Cells were then washed 3 × with PBS and resuspended in half the initial culture volume taken for washing. The Laurdan dye (Cat no. 40227, Sigma-Aldrich, Darmstadt, Germany) was added to the bacteria with a final concentration of 10 µM at room temperature in the dark. One hundred microliters of this dye/bacteria mixture was added to serially diluted peptides in a black, clear-bottom, 96-well plates (Cat no. 3904, Corning, NY, USA). After incubating in the dark for 1 h at room temperature, fluorescence intensity was measured using a spectrophotometer SpectraMax M2 (Molecular Devices, CA, USA) with excitation at 350 nm and a dual emission at 435 nm and 490 nm. The Laurdan GP was calculated using the formula GP = (I_435_ − I_490_)/(I_435_ + I_490_). 40 mM of benzyl alcohol as a membrane fluidizer, were used as a positive control.

### 4.8. S. Aureus Persisters Cell Generation and Time-Kill Assay

For the generation of antibiotic-induced persister cells, we followed an established protocol [42]. Twenty five ml of an *S. aureus* MW2 culture was grown to stationary phase and then treated with gentamicin at 20 μg/mL for 4 h for MRSA persister generation. The bacteria were washed with the same volume of phosphate-buffered saline (PBS) 3 times and then diluted with PBS to OD_600_ = 0.3 (~1 × 10^8^ CFU/mL). For killing the kinetics of the persisters, 1 mL of the cell suspension containing 1×, 2.5×, 5× and 10× MICs of C18 was added to the wells of a 2 mL deep well assay block (Cat no. 3960, Corning, NY, USA) and incubated at 37 °C, then shaken at 225 rpm. At specific times, 50 μL samples with different concentrations were diluted serially and inoculated on tryptic soy agar (TSA) (BD Difco, NJ, USA) plates. After the culture had been at 37 °C for 18 h, the colonies were counted. These experiments were also conducted in triplicate.

### 4.9. Plasmid Band Shift Assay

The methods in the reference were used to detect the ability of C18 binding DNA [43]. Briefly, the reaction of 20 μL containing 100 ng psGABO-SPE plasmid DNA (Addgene, MA, USA), polymyxin B, and C18 (1.95–500 μM) or ddH_2_O (control) was incubated in TE (Tris-EDTA) buffer (Invitrogen, CA, USA) at 37 °C for 2 h. The products were detected in 1% agarose gel electrophoresis, and plasmid migration was observed on a G: BOX Chemi XT4 gel doc system.

### 4.10. Prevention of S. aureus Static Biofilm Attachment

The ability of the peptides to prevent the initial bacterial attachment was done using an established protocol with minor modifications [44,45]. In short, high-density overnight cultures of *S. aureus* MW2 were grown in TSB media. Forty microliters of this culture were added to each well of the microtiter plates containing 50 µl of serially diluted 2 × peptide solution. Media containing bacteria and water were treated as a positive control, while only media with water served as the negative control. The plates were then incubated at 37 °C for 1 h. The media were carefully pipetted out, and the wells were washed with PBS (GIBCO, MD, USA) to remove loosely attached planktonic cells, followed by the addition of XTT [2,3-bis(2-methyloxy-4-nitro-5-sulfophenyl)-2H-tertazolium-5-carboxanilide]. Quantitation of the inhibition of biofilm attachment was done by assay following manufacture instructions with minor adjustments (ATCC, VA, USA). The calorimetric intensity of the XTT dye at 450 nm was measured by the SpectraMax M2 Multi-mode Microplate Reader (Molecular Devices, CA, USA). The percentage of biofilm growth was plotted, assuming 100% biofilm growth is achieved in the bacterial wells without peptide treatment.

### 4.11. Inhibition of S. aureus Biofilm Formation

The effectiveness of the peptides to inhibit the formation of biofilms was evaluated by following an established protocol with modifications [44,45]. In brief, exponential cultures of *S. aureus* MW2 were prepared in TSB from overnight inoculated cultures. Fifty μL of bacterial suspensions adjusted to OD_600_ ~0.03 in fresh TSB were incubated with 50 µl of serially diluted 2 × peptide solution in a flat-bottomed 96-well polystyrene microtiter plates (Cat No. 3903, Corning, NY, USA) for 48 h at 37 °C without shaking. Media containing bacteria and water was treated as a positive control while the media treated with water served as the negative control. After the incubation period, biofilm was quantified by the same method described above in the attachment experiment.

### 4.12. Quantitative Polymerase Chain Reaction (qPCR)

*S. aureus* MW2 culture was grown overnight in TSB to evaluate the effect of C18 on bacterial virulence genes. The exponential phase culture was harvested when the OD_600_ had reached ~0.4. The cells were washed with PBS and exposed to the half MIC concentration for 1 h. RNA was isolated using the RNeasy mini kit (Qiagen, Hilden, Germany) based on the manufacturer’s instructions. cDNA synthesis and quantitative reverse transcription (RT)-PCR were carried out as recommended by the manufacturer (Bio-Rad, CA, USA) using the primers listed in Appendix A. The qPCR cycling conditions were: 95 °C for 30 s; 40 cycles at 95 °C for 5 s; 55 °C for 30 s; finishing with a melt curve analysis from 65 to 95°C.

### 4.13. Hemolysis of Human Red Blood Cells (hRBCs)

The ability of the peptide to cause hemoglobin leakage was evaluated, as described in a previous study [41]. Human erythrocytes were purchased from Rockland Immunochemicals (Limerick, PA, USA), washed 3 × times in equal volume PBS, and resuspended as 4% hRBCs solution. One hundred microliters of the blood cells were added to 100 µl of the peptide solution in PBS serially diluted from 4–128 µg/mL in a 96-well microtiter plate. PBS and 1% Triton-X 100 were used as positive and negative controls. Upon adding the blood cells, the plate was incubated at 37 °C for 1 h and then centrifuged at 500× *g* for 5 min. One hundred microliters of the supernatant were transferred to a fresh 96-well plate, and absorbance was read at 540 nm. The hemolysis percentage was calculated considering 100% hemolysis caused by 1% Triton X-100 and 0% on PBS. The values were represented as the mean of duplicates. The following formula calculated the percentage of hemolysis: (A_540_ nm in the peptide solution − A_540_ nm in PBS)/(A_540_ nm of 1% Triton X-100 treated sample − A_540_ nm in PBS) × 100. Zero and 100% percentage hemolysis were determined in PBS and 0.1% Triton X-100, respectively.

### 4.14. Mammalian Cell Cytotoxicity Assays

HepG2 cells were used to assess the mammalian cell cytotoxicity of AMPs, as described previously [46,47]. Cells were grown in Dulbecco’s Modified Eagle Medium (DMEM) (Gibco, MD, USA) supplemented with 10% fetal bovine serum (FBS) (Gibco, MD, USA) and 1% penicillin/streptomycin (Gibco, MD, USA) and maintained at 37°C in 5% CO_2_. Cells were harvested and resuspended in a fresh medium, and 100 µL were distributed in a 96-well plate at 1 × 10^6^ cells/well. AMPs were serially diluted in serum and antibiotic-free DMEM added to the monolayer of the cells, and the plates were incubated at 37 °C in 5% CO_2_ for 24 h. At 4 h, before the end of the incubation period, 10 µL of 2-(4-iodophenyl)-3-(4-nitrophenyl)-5-(2, 4-disulfophenyl)-2H-tetrazolium (WST-1) solution (Roche, Mannheim, Germany) was added to each well. WST-1 reduction was monitored at 450 nm using a Vmax microplate reader. Assays were performed in triplicate, and the percentage of cell survival was calculated.

### 4.15. Galleria Mellonella In Vivo Assay

To study the protective effects of the peptide C18 in an in vivo model, we used an established wax moth model system [48]. *G. mellonella* larvae (Vanderhorst Wholesale, St. Mary’s, OH, USA) were distributed as a different group with randomly selected larvae (n = 16/group). The *S. aureus* MW2 bacteria were grown to exponential phase from overnight culture. Cells were then washed with PBS three times and finally suspended in PBS with an OD_600_ = 0.3. Experimental groups consisted of untouched (no injection), PBS (vehicle), bacterial infection group, and treatment group. Vancomycin was used as a positive control. Each of the groups were injected with 10 μL (2 × 10^6^ cells/mL) of the prepared bacteria, followed by peptide treatment (4–32 mg/kg doses were used) after 1 h. All injections were performed using a Hamilton syringe on the last left proleg. The larvae were incubated at 37 °C for 5 days, with live and dead counts being performed every 24 h.

### 4.16. Statistical Analysis

Student’s *t*-test was used for statistical analysis, and *p* < 0.05 was considered a significant difference. For the *G. mellonella* survival experiment, the Kaplan–Meier curve was plotted using GraphPad Prism Version 6.04 (GraphPad Software, La Jolla, CA). The same program was used for statistical analysis, and *p* < 0.05 was considered significant.

## Figures and Tables

**Figure 1 antibiotics-10-00036-f001:**
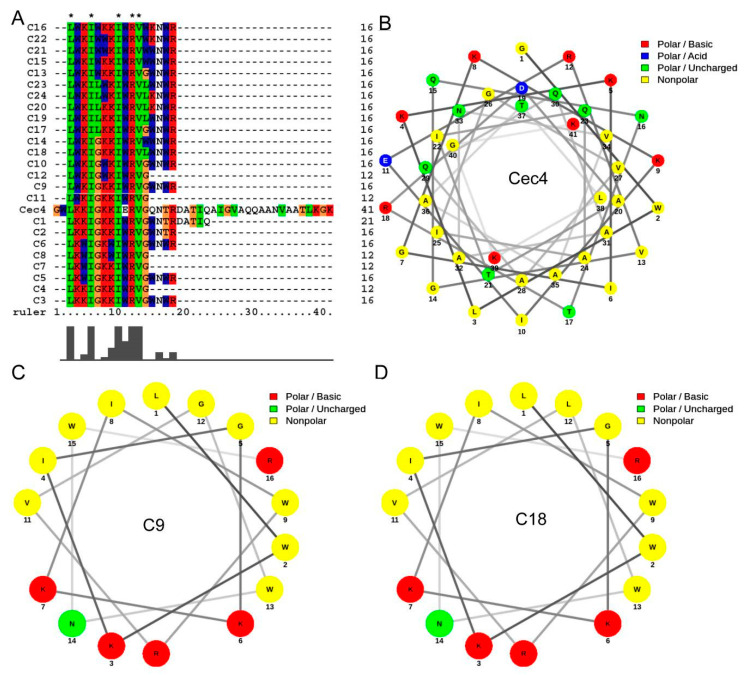
Small peptide sequences: alignment and helical wheels. (**A**) Sequenced alignment of Cec4-derived peptides with parental peptide. It shows the changes and relative positions of the amino acids of the derived peptide and the parental peptide. Multiple alignments were performed using the ClustalW program. The symbol (*) indicates that the aligned residues are identical. (**B**–**D**) The helical wheels of Cec4, C9 and C18 were shown respectively. The helical wheels were constructed with NetWheels (http://lbqp.unb.br/NetWheels/).

**Figure 2 antibiotics-10-00036-f002:**
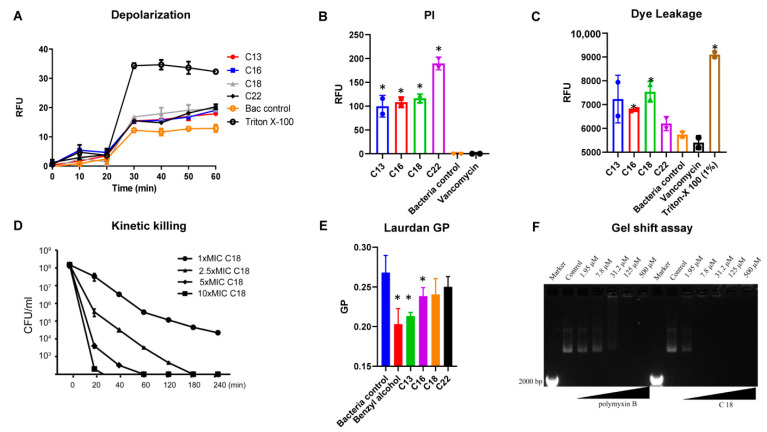
The interaction of Cec4-derived peptides with the cell membrane and DNA of *S. aureus* MW2 (**A**) Membrane depolarization of the bacterial surface upon treatment of peptides C13, C16, C18 and C22 at 32 µg/mL, accessed by the DISC3 (5) dye. Triton X-100 (1%) was used as a control. (**B**) Membrane permeation of propidium iodide as a pore formation mechanism marker. The percentage of permeabilization is plotted to compare with 1% Triton X-100 at a peptide concentration of 32 µg/mL. (**C**) Dye leakage of rhodamine-BF from DOPC: cholesterol liposomes. (**D**) Kinetic killing of *S aureus* persisters (**E**) Laurdan GP plots measuring membrane fluidity change upon the interactions between the peptides and (**F**) The plasmid shift caused by the peptide DNA binding. * *p* < 0.05, Student’s t-test was compared with the control.

**Figure 3 antibiotics-10-00036-f003:**
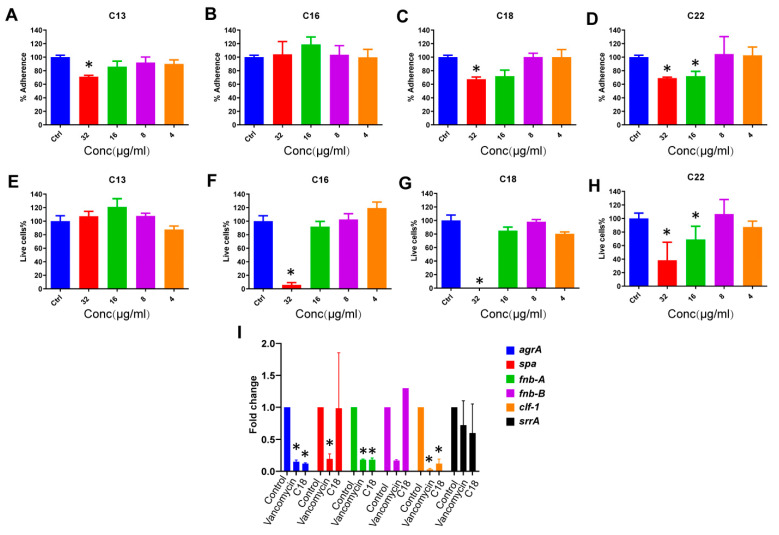
The antibiofilm effects of Cec4-derived peptides. (**A**–**D**) The potential of peptides to inhibit *S. aureus* MW2 biofilm attachments. (**E**–**H**) The inhibition of *S. aureus* MW2 biofilm formation. (**I**) *S. aureus* MW2 cells were treated with C18 (2 μg/mL) for 4 h. Exposure to C18 was associated with down-regulation of major virulence regulator genes, including agrA, fnb-A, and clf-1. Data represent the mean ± SD (n = 3). * *p* < 0.05, Student’s t-test compared with PBS control.

**Figure 4 antibiotics-10-00036-f004:**
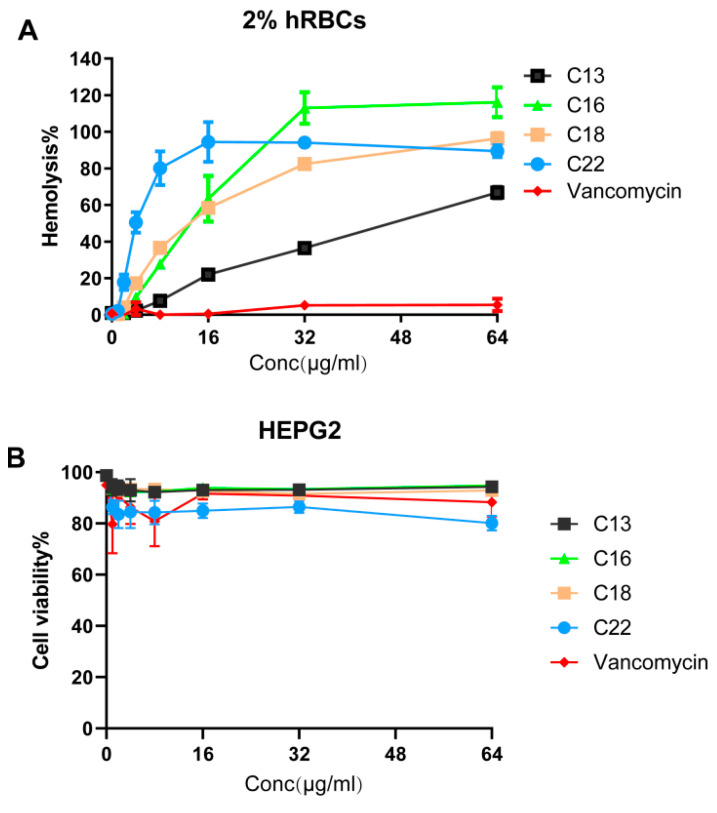
The hemolytic and cytotoxic effects of Cec4-derived peptides. (**A**) The hemolysis of human red blood cells (hRBCs) indicated that the HL_50_ of the C13, C16, C18 and C22 peptides were 48, 16, 16, and 4 µg/mL, respectively. Vancomycin was used as a negative control. (**B**) Cellular toxicity of the peptide against liver-derived HepG2 cell lines showed that the LD_50_ of all tested peptides were >64 µg/mL.

**Figure 5 antibiotics-10-00036-f005:**
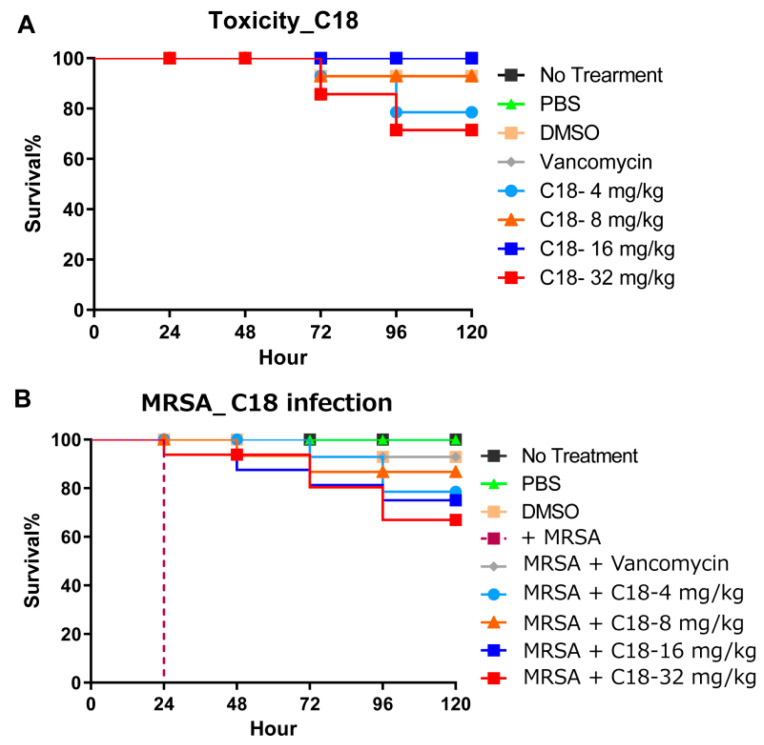
The in vivo activity of the designed peptide in a *G. mellonella* model. (**A**) The toxicity of the peptide in the whole animal model. More than 70% of larvae survived the toxicity even at 32 mg/kg until 120 h (*p* < 0.05). (**B**) The protection of the *G. mellonella* larvae by the peptide and the antibiotic post-MRSA infection. Similar concentrations of the agents were used to compare the toxicity with the actual protection effects. Larvae treated at 4 mg/kg showed more than 80% survival at 120 h post-infection (*p* < 0.05).

**Table 1 antibiotics-10-00036-t001:** Minimal growth inhibition concentrations (MICs) of Cec4-derived antimicrobial peptides (AMPs) against tested microorganisms (µg/mL, μM).

Peptides	Sequences	*E. faecium*	*S. aureus (ATCC29231)*	*S. aureus MW2*	*K. pneumoniae*	*A. baumannii*	*P. aeruginosa*	*E. aerogenes*	*C. albicans*
		G+			G−				Fungus
C1	LKKIGKKIWRVGWNTRDATIQ	>128/50.96	>128/50.96	>128/50.96	>128/50.96	>128/50.96	>128/50.96	>128/50.96	>128/50.96
C2	LKKIGKKIWRVGWNTR	>128/64.53	>128/64.53	>128/64.53	>128/64.53	128/64.53	>128/64.53	>128/64.53	>128/64.53
C3	LKKIGKKIWRVGWNWR	>128/61.87	>128/61.87	128/61.87	64/32.27	32/16.13	128/61.87	>128/61.87	128/61.87
C4	LKKIGKKIWRVG	>128/89.77	>128/89.77	>128/89.77	>128/89.77	>128/89.77	>128/89.77	>128/89.77	>128/89.77
C5	LKWIGKKIWRVGWNWR	>128/60.19	128/60.19	64/30.10	128/60.19	32/15.05	128/60.19	>128/60.19	128/60.19
C6	LKWIGKWIWRVGWNWR	>128/58.59	>128/58.59	64/29.30	>128/58.59	128/58.59	>128/58.59	>128/58.59	>128/58.59
C7	LKWIGKKIWRVG	>128/86.26	>128/86.26	>128/86.26	>128/86.26	>128/86.26	>128/86.26	>128/86.26	>128/86.26
C8	LKWIGKWIWRVG	>128/83.01	>128/83.01	64/41.51	>128/83.01	>128/83.01	>128/83.01	>128/83.01	>128/83.01
C9	LWKIGKKIWRVGWNWR	>128/60.19	64/30.10	64/30.10	64/30.10	16/7.25	64/30.10	128/60.19	128/60.19
C10	LWKIGWKIWRVGWNWR	>128/58.59	32/14.65	32/14.65	128/58.59	32/14.65	64/29.30	128/58.59	64/29.30
C11	LWKIGKKIWRVG	>128/86.26	>128/86.26	>128/86.26	>128/86.26	>128/86.26	>128/86.26	>128/86.26	>128/86.26
C12	LWKIGWKIWRVG	>128/83.01	64/41.51	64/41.51	64/41.51	32/20.75	128/83.01	128/83.01	128/83.01
C13	LWKIWKKIWRVGWNWR	>128/56.74	16/7.09	32/14.19	32/14.19	32/14.19	32/14.19	64/28.37	32/14.19
C14	LWKIGKKIWRVWWNWR	>128/56.74	16/7.09	32/14.19	32/14.19	32/14.19	32/14.19	64/28.37	32/14.19
C15	LWKIWKKIWRVWWNWR	128/53.67	32/13.14	64/26.84	128/53.67	128/53.67	64/26.84	128/53.67	128/53.67
C16	LWKIWKKIWRVWKNWR	>128/55.01	8/3.44	32/13.75	64/27.50	64/27.50	32/13.75	32/13.75	32/13.75
C17	LWKILKKIWRVGWNWR	>128/58.64	16/7.33	32/14.66	64/29.32	64/29.32	64/29.32	128/58.64	64/29.32
C18	LWKIGKKIWRVLWNWR	128/58.64	4/1.83	4/1.83	16/7.33	16/7.33	32/14.66	16/7.33	16/7.33
C19	LWKILKKIWRVLWNWR	>128/57.17	32/14.29	64/28.59	128/57.17	128/57.17	128/57.17	128/57.17	64/28.59
C20	LWKILKKIWRVLKNWR	>128/58.70	32/14.67	32/14.67	64/29.35	128/58.70	32/14.67	64/29.35	32/14.67
C21	LWKIWWKIWRVWWNWR	>128/52.40	128/52.40	128/52.40	>128/52.40	128/52.40	>128/52.40	>128/52.40	>128/52.40
C22	LWKIWWKIWRVWKNWR	>128/53.67	8/3.35	32/13.42	128/53.67	>128/53.67	32/13.42	128/53.67	64/26.84
C23	LWKILWKIWRVLWNWR	>128/55.73	128/55.73	128/55.73	>128/55.73	128/55.73	>128/55.73	>128/55.73	128/55.73
C24	LWKILWKIWRVLKNWR	128/57.2	64/28.59	128/57.2	128/57.2	>128/57.2	128/57.2	128/57.2	>128/57.2

**Table 2 antibiotics-10-00036-t002:** The effects of salts and serum on peptide activity against *S. aureus* MW2, MIC (μg/mL).

Peptides	MHB	NaCl (150 mM)	CaCl_2_ (2 mM)	Serum (5%)
C13	16	32	16	128
C18	4	4	4	128
C22	32	32	32	>128
vancomycin	2	2	2	2

**Table 3 antibiotics-10-00036-t003:** Synergy analysis of C18 with clinical antibacterial agents.

SL. No.	Antibacterial Agents	FICI
1	daptomycin	0.313
2	vancomycin	0.625
3	gentamicin	1.25
4	oxacillin	0.75
5	ciprofloxacin	0.75

## Data Availability

Data is contained within the article.

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
