# Peer review of "Novel Cecropin-4 Derived Peptides against Methicillin-Resistant Staphylococcus aureus"

_antibiotics, 2021, doi:10.3390/antibiotics10010036_

Round 1
Reviewer 1 Report
- General comment
In the manuscript, it was shown that antimicrobial peptide C18 derived from cecropin C4 was an attractive therapeutic candidate to be developed to treat severe MRSA infections. The manuscript represents an advance in the understanding the utilization of modified peptide derived from already-known antimicrobial peptide as novel therapeutic candidate.
2. Major revision
1) As the molecular weights of the peptides (C1~C22) were slightly different each other, it is strongly recommended to explain and discuss the data using two numerical values, µM and µg/mL, in the manuscript, similarly to line 219~220.
2) Figure 4B and line 272~276
- a) It is essential to explain and discuss the survival % of +SA and vertical broken line shown in Figure 4B, comparing with those of SA+C18-4~32 mg/kg.
- b) It is essential to check the sentence of line 272~273 “All mellonella larvae infected with MRSA were alive at 120 hours”.
- c) It is essential to revise “SA” to “MRSA” in Figure 4B.
3) Table SII
It is recommended to show Table SII as Table 1 in the manuscript, including the amino acid sequence of Cec4. In addition, it is recommended to classify the amino acid sequence of Cec4 derived peptides (C1~C24) by color, similarly to Figure 2 of Ref.15 in the reference.
Author Response
Please see the attachement

Reviewer 2 Report
In this study, the authors designed a series of antimicrobial peptides derived from the N-terminal helix of the cecropin4 (Cec4). The designed peptide C18 (LWKIGKKIWRVLWNWR) was found to exhibit broader antibacterial activity against drug-resistant S. aureus strains (4 µg/ml) than the parent peptide. In addition, C18 displayed synergistic effects in combination with daptomycin against S. aureus strain. Moreover, the authors also proved that C18 peptide killed bacteria through membrane disruption by membrane depolarized assay, propidium iodide-based membrane permeability, membrane fluidity assay, time-killing assay. In addition, the C18 peptide also exhibits DNA binding and anti-biofilm activities. In vivo assays showed that C18 is significantly efficacious in the Galleria mellonella (was moth) model. However, this manuscript lacks discussion about the structure-function relationship of the designed peptides.
Major comments
- Sequence number needs to be consistent. For example, Lys2/3 in line 125 is numbered according to peptide C1, but Gly7 and 14 in line 145 is numbered according to peptide Cec4. Sequence of each peptide should be listed in Table 1.
- How do the authors compare hydrophobicity and charge between designed peptides? They should provide more detail explanations.
- The function of each virulence genes used in this study should be explained.
- In figure 2I, the legend of blue bar should be corrected as “agr A”.
- Supplementary materials PDF.1 & Table S1 are not provided.
- (page 3, line 150) Peptide C17 should be obtained by reverse substitution of Leu5 with Gly from peptide C9 according to Table S2.
- Information about helical wheel, hydrophobicity and hydrophobic moment of the peptides should be provided.
- MICs for these peptides in 5% serum were higher than HL50%, these peptides may not be suitable for systemic therapy. Another suitable cytotoxicity assay should be used.
- Toxicity and efficacy studies in a mellonella model demonstrated that survival rate was more than 80% with the dose of 8mg/kg, with or without S.A. at 120 hrs post administration. However, the survival rate was about 70% with the dose of 4mg/kg, with or without S.A. at 120 hr post administration. The authors should provide an explanation for this difference.
Round 2
Reviewer 2 Report
The authors have answered all of my questions.